# On the Uncertainty in Prior-Data Fitted Network Pretraining

**Manuel Hülskamp** [1]  **Julius Kobialka** [1 2]  **Emanuel Sommer** [1 2]  **David Rügamer** [1 2]

## Abstract

Prior-data fitted networks (PFNs) have recently emerged as a new paradigm for supervised machine learning by approximating Bayesian posterior predictive distributions via in-context learning. While leading to state-of-the-art predictive performance across a wide range of benchmarks, quantifying the approach's uncertainty is challenging. Using an empirical risk minimization perspective, we characterize possible sources of uncertainty in PFN predictions and empirically show that certain uncertainties cannot be reduced by simply scaling the model size or pretraining. To explicitly address these overlooked sources of uncertainty, we study deep ensembles of PFNs and formulate a Bayesian neural network version of PFNs for which we obtain samples from an approximate posterior via Markov chain Monte Carlo.

## 1. Introduction

Prior-data fitted networks (PFNs) have emerged as powerful foundation models for tabular data (Müller et al., 2021; Hollmann et al., 2025). By learning to approximate Bayesian posterior predictive distributions via in-context learning (Brown et al., 2020) during pretraining, they perform inference on new downstream datasets in a single forward pass. Given the success of these models, several works have proposed strategies or alternative formulations to improve predictive performance, scale to larger datasets, or transfer the approach to other tasks (Qu et al., 2025; 2026; Zhang et al., 2025a; Breejen et al., 2024; Thomas et al., 2024; Xu et al., 2024; Liu & Ye, 2025; Ma et al., 2025; Robertson et al., 2025; Balazadeh et al., 2025). However, uncertainty quantification for PFNs has received relatively little attention (Nagler & Rügamer, 2025; Fortini et al., 2026), and

most importantly, existing work has exclusively focused on specific uncertainties at inference time, leaving the pretraining process untouched. This raises the question: *What uncertainties exist in the pretraining process of PFNs and how can they be quantified?*

To address these questions, our contributions are:

1. A conceptual framework analysing possible sources of uncertainty in PFNs via a formulation of PFN pretraining as empirical risk minimization, uncovering two so far overlooked sources of uncertainty (Section 2.2).

2. A scaling analysis showing that one such uncertainty, optimization uncertainty, *persists* even when scaling both the number of pretraining datasets $M$ and model size (Section 3.1).

3. BDE-PFNs: A training-time ensembling approach for PFNs addressing optimization uncertainty, instantiated via deep ensembles followed by MCMC-based posterior sampling, which yields improvements over individual models (Section 3.2).

A more detailed discussion of related work is provided in Section A, with additional methodological details in Section D and further results in Sections C, E and H.

## 2. Decomposing PFN Pretraining Uncertainty

### 2.1. Setup

Given a dataset of *iid* samples $\mathcal{D}_n = \{(Y_i, \mathbf{X}_i)\}_{i=1}^n$ with features $\mathbf{X} \in \mathcal{X} \subseteq \mathbb{R}^d$ and labels $Y \in \mathcal{Y}$ from an underlying distribution $p_0$, supervised learning aims to model the conditional distribution $p_0(y \mid \mathbf{x})$. Following Nagler (2023), PFNs approach this by assuming a prior over data-generating mechanisms $\Pi$ and, for each dataset sample size $n$, approximating $p_0(y \mid \mathbf{x})$ via the posterior predictive (PPD) under this prior as

$$\pi_\Pi(y \mid \mathbf{x}, \mathcal{D}_n) = \int p(y \mid \mathbf{x}) d\Pi(p \mid \mathcal{D}_n). \quad (1)$$

This PPD is then again approximated by a model $q_{\boldsymbol{\theta}}$ with parameters $\boldsymbol{\theta} \in \Theta$ and learned by a neural network via

$$\boldsymbol{\theta}^* = \arg\max_{\boldsymbol{\theta} \in \Theta} \underbrace{\mathbb{E}_\Psi \mathbb{E}_\Pi [\log q_{\boldsymbol{\theta}}(Y \mid \mathbf{X}, \mathcal{D}_n)]}_{\mathcal{L}(\boldsymbol{\theta})}, \quad (2)$$

[1]Department of Statistics, LMU Munich, Munich, Germany [2]Munich Center for Machine Learning, Munich, Germany. Correspondence to: Manuel Hülskamp <m.huelskamp@campus.lmu.de>.

*Proceedings of the 2$^{nd}$ ICML Workshop on Foundation Models for Structured Data*, Seoul, South Korea. 2026. Copyright 2026 by the author(s).

where $\Psi$ is a distribution over the sample size of the datasets $\mathcal{D}_n$. As Müller et al. (2021) showed, this is equivalent to minimizing the KL-divergence $D_{\mathrm{KL}}$ between $q_{\boldsymbol{\theta}}(\cdot \mid \mathbf{X}, \mathcal{D}_n)$ and $\pi_\Pi(\cdot \mid \mathbf{X}, \mathcal{D}_n)$. To learn a concrete model, Equation (2) is once more approximated via classical empirical risk minimization over $M$ synthetic training datasets drawn from the prior $\Pi$ via

$$\hat{\boldsymbol{\theta}} = \arg\min_{\boldsymbol{\theta} \in \Theta} \underbrace{-\frac{1}{M} \sum_{j=1}^{M} \log q_{\boldsymbol{\theta}}(y_j \mid \mathbf{x}_j, \mathcal{D}^{(j)})}_{=: \mathcal{L}_{\mathrm{emp}}(\boldsymbol{\theta})}, \quad (3)$$

which is optimized using first-order stochastic gradient optimizers. In the following, we will show that this approach can introduce uncertainty at three distinct levels: *Amortization uncertainty*, *context uncertainty*, and *prior uncertainty*.

## 2.2. Sources of Uncertainty in PFNs

For a given test feature $\mathbf{x}^*$ and a context dataset of size $n$, the *deployment error* of any model $q_{\hat{\boldsymbol{\theta}}}(\cdot \mid \mathbf{x}^*, \mathcal{D}_n)$, as measured by the KL-divergence to $p_0$ is given by

$$\begin{aligned} D_{\mathrm{KL}}\left(p_0(\cdot \mid \mathbf{x}^*) \,\|\, q_{\hat{\boldsymbol{\theta}}}(\cdot \mid \mathbf{x}^*, \mathcal{D}_n)\right) = \\ \underbrace{\mathbb{E}\left[\log \pi_\Pi(Y \mid \mathbf{x}^*, \mathcal{D}_n)\right] - \mathbb{E}\left[\log q_{\hat{\boldsymbol{\theta}}}(Y \mid \mathbf{x}^*, \mathcal{D}_n)\right]}_{\text{Amortization gap}} \\ + \underbrace{\mathbb{E}\left[\log p^*(Y \mid \mathbf{x}^*)\right] - \mathbb{E}\left[\log \pi_\Pi(Y \mid \mathbf{x}^*, \mathcal{D}_n)\right]}_{\text{Context gap}} \\ + \underbrace{\mathbb{E}\left[\log p_0(Y \mid \mathbf{x}^*)\right] - \mathbb{E}\left[\log p^*(Y \mid \mathbf{x}^*)\right]}_{\text{Prior error}}, \quad (4) \end{aligned}$$

where $p^* = \arg\min_{p \in \mathcal{P}} D_{\mathrm{KL}}(p_0 \,\|\, p)$ is the KL-optimal approximation of $p_0$ within the support of the prior $\Pi$, denoted by $\mathcal{P} = \{p : \Pi(p) > 0\}$. The first two terms are differences of cross-entropies and need not be non-negative pointwise.

By Nagler (2023, Theorem 3.1), $\pi_\Pi(\cdot \mid \mathbf{x}, \mathcal{D}_n) \to p^*$ a.s. as $n \to \infty$. If $p_0 \in \mathcal{P}$ holds, then $p^* = p_0$, $D_{\mathrm{KL}}(p_0 \,\|\, p^*) = 0$, and the above error reduces to

$$\begin{aligned} D_{\mathrm{KL}}\left(p_0(\cdot \mid \mathbf{x}^*) \,\|\, q_{\hat{\boldsymbol{\theta}}}(\cdot \mid \mathbf{x}^*, \mathcal{D}_n)\right) = \\ \underbrace{\mathbb{E}\left[\log \pi_\Pi(Y \mid \mathbf{x}^*, \mathcal{D}_n)\right] - \mathbb{E}\left[\log q_{\hat{\boldsymbol{\theta}}}(Y \mid \mathbf{x}^*, \mathcal{D}_n)\right]}_{\text{Amortization gap}} \\ + \underbrace{\mathbb{E}\left[\log p_0(Y \mid \mathbf{x}^*)\right] - \mathbb{E}\left[\log \pi_\Pi(Y \mid \mathbf{x}^*, \mathcal{D}_n)\right]}_{\text{Context gap}}, \quad (5) \end{aligned}$$

and taken in expectation over the training prior, the two terms do become non-negative KL-divergences. However, if $p_0 \notin \mathcal{P}$, the prior error does not vanish. This highlights two things: First, in both cases the amortization gap can remain non-zero, and second, in case of prior misspecification, the three sources, and subsequently uncertainty for any given $q_{\hat{\boldsymbol{\theta}}}$ are likely impossible to disentangle and quantify from the predictive distribution $q_{\hat{\boldsymbol{\theta}}}$ alone. This warrants specific instruments to decompose uncertainty of PFNs.

## 2.3. Error Components

While this paper focuses on the amortization uncertainty, we will briefly discuss the two other types as well.

**Prior and context uncertainty** The prior error is present if the prior $\Pi$ is misspecified for a concrete dataset $\mathcal{D}_n$. While this issue, also known as the *simulation gap*, has been studied in the amortized and simulation-based inference literature (see, e.g., Schmitt et al., 2023), we are not aware of any investigation for PFNs specifically. Measuring the *severity* of this uncertainty for a given model and quantifying the impact of it on predictive performance is an avenue for further research. In contrast, the context uncertainty is the object being studied by Nagler & Rügamer (2025); Fortini et al. (2026): For a small, finite $n$, any posterior-predictive must be uncertain, as $n$ increases the posterior $\Pi(p \mid \mathcal{D}_n)$, concentrates on $p^*$. Both approaches rely on martingale posteriors (Fong et al., 2023) to study this uncertainty and separate it from aleatoric uncertainty (variance in the conditional law $Y \mid \mathbf{x}^*$).

**Amortization uncertainty** We now turn to our main objective, the amortization uncertainty. It arises during pretraining of $q_{\hat{\boldsymbol{\theta}}}$ when the network is not able to fully amortize the target function $\pi_\Pi$ due to limited capacity or suboptimal optimization. While the amortization gap could be seen as an unknown but fixed property of any trained model, formulating it in terms of a concrete distribution $p_0$ makes it practically more interesting and easier to quantify, answering: *How consequential is the amortization gap for my given task?* To do so, the amortization gap can be further decomposed into the classical three components of empirical risk minimization:

$$\begin{aligned} \mathbb{E}\left[\log \pi_\Pi(Y \mid \mathbf{x}^*, \mathcal{D}_n)\right] - \mathbb{E}\left[\log q_{\hat{\boldsymbol{\theta}}}(Y \mid \mathbf{x}^*, \mathcal{D}_n)\right] = \\ \underbrace{\left[\mathbb{E}\left[\log q_{\tilde{\boldsymbol{\theta}}_M}(Y \mid \mathbf{x}^*, \mathcal{D}_n)\right] - \mathbb{E}\left[\log q_{\hat{\boldsymbol{\theta}}}(Y \mid \mathbf{x}^*, \mathcal{D}_n)\right]\right]}_{\text{optimization error}} \\ + \underbrace{\left[\mathbb{E}\left[\log q_{\boldsymbol{\theta}^*}(Y \mid \mathbf{x}^*, \mathcal{D}_n)\right] - \mathbb{E}\left[\log q_{\tilde{\boldsymbol{\theta}}_M}(Y \mid \mathbf{x}^*, \mathcal{D}_n)\right]\right]}_{\text{estimation error}} \\ + \underbrace{\left[\mathbb{E}\left[\log \pi_\Pi(Y \mid \mathbf{x}^*, \mathcal{D}_n)\right] - \mathbb{E}\left[\log q_{\boldsymbol{\theta}^*}(Y \mid \mathbf{x}^*, \mathcal{D}_n)\right]\right]}_{\text{approximation error}}, \end{aligned}$$
$$(6)$$

where $\tilde{\boldsymbol{\theta}}_M$ minimizes the empirical risk (Equation (3)) and $\boldsymbol{\theta}^*$ minimizes the population risk (Equation (2)) within the model class. The **estimation error** is what classical statistical learning theory ties to the number of training samples $M$ and expects to decrease at $\sqrt{M}$-rate (Nagler, 2023). The **approximation error** is a property of the hypothesis class $\Theta$ and therefore can only be decreased by enlarging it. **Optimization error** is a consequence of the chosen optimization method and, in general, is not guaranteed to decrease with

a scaling in model size or training duration, as PFNs are (overparameterized) neural networks optimized via stochastic gradient methods on a non-convex empirical risk. In the following section, we propose how to quantify the uncertainties induced by the components of the amortization gap to subsequently study their empirical behaviour for concrete PFNs.

### 2.4. Quantifying Amortization Uncertainty

The three components in the amortization gap show different dependencies on the two scaling axes of model-size and pretraining scale $M$, and we quantify them accordingly. The **estimation error** can be studied along the number of pretraining datasets $M$, and we propose to estimate it by training $K$ independent models that share the same optimizer settings but differ in seeds and training data at each of several meta-training scales $M_1 < \cdots < M_L$. We then compute

$$\widehat{EE}(M) = \frac{1}{K} \sum_{k=1}^{K} \mathcal{L}_{\text{test}}(\boldsymbol{\theta}_M^{(k)}) - \mathcal{L}_{\text{test}}(\boldsymbol{\theta}_{\text{ref}}), \quad (7)$$

where $\boldsymbol{\theta}_{\text{ref}}$ is our highest-$M$ reference checkpoint with $M_{\text{ref}} \gg M_L$. Averaging over the $K$ warmstarts cancels the zero-mean optimization noise that would otherwise contaminate any individual run, isolating the estimation contribution. We choose this biased estimator as a ground-truth minimizer of the population objective is not tractable. We then fit a linear regression for $\log \widehat{EE}(M) = \log C - \alpha \log M$ to recover the rate $\alpha$. **Optimization uncertainty** (OU), in contrast, is not defined relative to $M$ but measures the functional spread across $K$ independently trained models $\{\boldsymbol{\theta}^{(k)}\}_{k=1}^{K}$ that share the same $M$, model size, and optimizer settings but differ in random seed and training data. To measure the optimization error, we propose to track four complementary metrics: the Jensen–Shannon divergence $OU_{\text{JSD}}$, the pairwise argmax disagreement rate $OU_{\text{dis}}$, the across-warmstart standard deviation of predicted class probabilities $OU_{\text{std}}$, and the across-warmstart standard deviation of test NLL $OU_{\text{NLL}}$ (as defined in Section D). To test whether OU is driven to zero by scaling, we report it *along both axes*: as a function of $M$ (data axis) and at three model sizes (capacity axis). **Approximation error** can only be studied along the capacity axis but is difficult to study directly, as it requires access to, or at least good estimates of, $q_{\boldsymbol{\theta}^*}$ for a range of hypothesis classes with increaseing capacity and $\pi_\Pi$, which is not analytically tractable for the priors $\Pi$ used in actual PFNs. We thus defer a more detailed analysis to future work.

### 2.5. Bayesian Deep Ensemble PFNs

As amortization uncertainty can be considered a pretraining uncertainty in the weight space of the neural network

parametrizing the PFN, we propose to adopt a probabilistic perspective on PFN pretraining and approximate a posterior over network weights. We refer to the resulting predictors as *Bayesian Deep Ensemble PFNs* (BDE-PFNs) by combining BDEs (see, e.g., Sommer et al., 2025) with PFNs. Placing a Gaussian prior $p(\boldsymbol{\theta}) = \mathcal{N}(0, \sigma_{\boldsymbol{\theta}}^2 I)$ on all weights $\boldsymbol{\theta}$ and combining it with the PFN likelihood from Equation (3) yields $p(\boldsymbol{\theta} \mid \mathcal{T}_M) \propto \exp(-[M \cdot \mathcal{L}_{\text{emp}}(\boldsymbol{\theta}) - \log p(\boldsymbol{\theta})])$ with $\mathcal{T}_M = \{((y_j, \mathbf{x}_j), D^{(j)})\}$ denoting the collection of all pretraining datasets. Since $M$ can be made arbitrarily large, this posterior collapses onto minimizers of $\mathcal{L}(\boldsymbol{\theta})$ for $M \to \infty$. To retain meaningful uncertainty, we instead target a *tempered posterior* (Wenzel et al., 2020)

$$p_\beta(\boldsymbol{\theta} \mid \mathcal{T}_M) \propto \exp\left(-\left[\beta \cdot \mathcal{L}_{\text{emp}}(\boldsymbol{\theta}) - \log p(\boldsymbol{\theta})\right]\right), \quad (8)$$

where $\beta > 0$ decouples posterior concentration from $M$. Predictions are obtained by averaging over approximate posterior samples,

$$\pi_{\text{BDE}}(y \mid x, \mathcal{D}_n) = \frac{1}{S} \sum_{s=1}^{S} q_{\boldsymbol{\theta}^{(s)}}(y \mid x, \mathcal{D}_n). \quad (9)$$

We consider two strategies to obtain such samples. Deep ensembles (DEs; Lakshminarayanan et al., 2017) treat independently trained warmstarts as approximate draws from distinct local minima and provide a simple, robust baseline. Stochastic gradient MCMC methods sample directly from the tempered posterior; we use scale-adapted SGHMC (Chen et al., 2014; Springenberg et al., 2016), pSMILE (Sommer et al., 2026), and cyclical SGLD (Zhang et al., 2020), initializing each chain from a warm-started checkpoint (Sommer et al., 2024). Both approaches yield collections of parameter samples $\{\boldsymbol{\theta}^{(s)}\}_{s=1}^{S}$ that are combined via Equation (9). Implementation details and hyperparameters for all three methods are detailed in Section G.

## 3. Experimental Study

**Setup**   We study two PFN architectures reimplemented in JAX at three scales: *NanoTabPFN* (Pfefferle et al., 2025) ($\sim$2M /5M/10M params), and *NanoTabICL* (Qu et al., 2026) ($\sim$2M/7M/14M params). Both are pretrained on synthetic classification tasks from the TabICL prior (Qu et al., 2025) with up to 500 rows, 50 features, and 5 classes. Warmstarts use AdamW with cosine annealing and are trained until *convergence*. Every warmstart and sampling chain uses disjoint training data. For small-scale BDEs, we use $K{=}8$ warmstarts for the uncertainty decomposition and chain initializations, $K{=}4$ for medium scale, and $K{=}2$ for the large scale.

**Evaluation and methods compared**   We report classification results using accuracy (ACC), ROC AUC, and log pointwise predictive density (LPPD) on a large synthetic

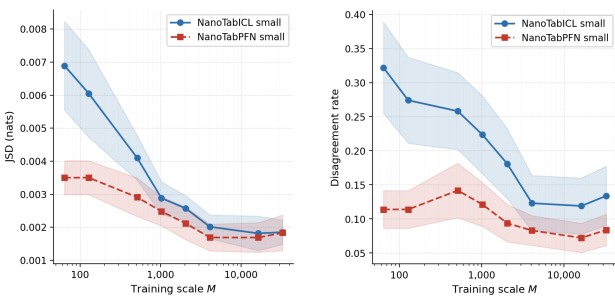

*Figure 1.* Optimization uncertainty as a function of meta-training scale $M$ for NanoTabICL and NanoTabPFN (small) evaluated on TabArena datasets. **Left**: Jensen–Shannon divergence. **Right**: Pairwise disagreement rate.

*Table 1.* Optimization uncertainty on TabArena datasets. Higher values indicate greater functional disagreement.

| Model | $OU_{\mathrm{JSD}}$ | $OU_{\mathrm{Dis}}$ | $OU_{\mathrm{Var}}$ | $OU_{\mathrm{NLL}}$ |
|---|---|---|---|---|
| NanoTabICL small | 0.0011 | 0.0966 | 0.0200 | 0.0106 |
| NanoTabICL medium | 0.0010 | 0.0595 | 0.0200 | 0.0098 |
| NanoTabICL large | 0.0006 | 0.0583 | — | — |
| NanoTabPFN small | 0.0013 | 0.0597 | 0.0221 | 0.0061 |
| NanoTabPFN medium | 0.0010 | 0.0637 | 0.0197 | 0.0057 |
| NanoTabPFN large | 0.0007 | 0.0650 | — | — |

test set drawn from the TabICL prior and on datasets drawn from a recent benchmark, namely TabArena (Erickson et al., 2025). We compare the average single warmstart (DE-(1) avg.), the deep ensemble (DE-($K$)), BDE-PFN posterior averages obtained via MCMC-Sampling, and the official pretrained TabPFNv2/TabICLv2 baselines.

### 3.1. Scaling Behaviour Results

We study how the two measurable components of Equation (6) respond to scaling along two axes: The number of meta-training datasets $M$ and the model size.

**Scaling in $M$** At fixed small model size, we compute (i) the estimation-error estimator $\widehat{EE}(M)$ of Equation (7) and (ii) the four OU metrics across the $K$ warmstarts at each scale. Figure 6 (App.) shows the estimation error scaling, Figure 1 presents the optimization uncertainty. The two components behave strikingly differently. $\widehat{EE}(M)$ decays at the theoretically expected $\sqrt{M}$ rate: the ordinary least squares fit yields $\alpha \approx 0.50$ (nanoTabICL small) and $\alpha \approx 0.44$ (nanoTabPFN small; Section E), both close to theoretical statement $\alpha = 0.5$. The four OU metrics, by contrast, decrease only initially over the scanned range and flatten out well before it ends, suggesting that training on substantially more synthetic data would not eliminate OU.

**Scaling in Model Size** Table 1 presents the optimization uncertainty metrics at three model sizes on the TabArena datasets, Table 2 (App.) for the synthetic test set. Scaling from $\sim$2M to $\sim$10$-$14M parameters roughly halves the JSD, indicating that the distribution over predicted class probabilities becomes somewhat sharper at larger capacity. The pairwise disagreement rate, which tracks whether two warmstarts produce the same argmax, tells a different story: It remains essentially unchanged on the synthetic test set and, on TabArena, does not shrink at all, even disagreeing slightly more often at a larger scale. Averages additionally conceal a heavy-tailed distribution across individual TabArena datasets: The maximum pairwise disagreement rate reaches 23.8% (nanoTabPFN) and 54.9% (nanoTabICL); cf. Figure 5 (App.). Together with the data-axis findings, this suggests that *neither scaling $M$ nor scaling model capacity drives optimization uncertainty to zero*: OU is a persistent property of PFN pretraining that requires dedicated mitigation strategies.

### 3.2. Bayesian Deep Ensembles

Training-time ensembles are the direct response to the uncertainty we have uncovered. The DEs over $K$ independent warmstarts consistently improve over the average single warmstart, presenting a pragmatic way to average over plausible training outcomes (see Figure 3 and Table 4). Furthermore, the improvement is not only over the average single warmstart, but also over the best single ensemble member, picked via the lowest validation loss, as Table 3 shows using the TabArena-light evaluation protocol. Interestingly, the MCMC-based approaches, grounding the ensemble average of Equation (9) more firmly in Bayesian theory, show tiny to no improvements over the DEs (Figure 4).

## 4. Discussion & Conclusion

Optimization uncertainty is the dominant component of epistemic uncertainty in pretraining for PFNs and invisible to inference-time methods: It can only be addressed during pretraining and is a persistent component of excess risk that likely does *not* vanish with the usual scaling levers: Estimation error decays at the theoretical $\sqrt{M}$ rate and becomes negligible at practical scales as PFNs are trained on purely synthetic data, while OU is flatter in $M$ and is not driven to zero by capacity. This establishes it as a distinct scaling axis orthogonal to, and complementary with, the other described sources of uncertainty during inference time for a concrete dataset $\mathcal{D}_n$. Training-time ensembles are the natural response, with DEs marginalizing over a substantial fraction of this uncertainty. Interestingly, more principled Bayesian approaches do not yet show substantial improvements over DEs. Limitations and future work are discussed in Section B.

## Acknowledgments

This paper is supported by the DAAD programme Konrad Zuse Schools of Excellence in Artificial Intelligence, sponsored by the Federal Ministry of Research, Technology and Space.

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

# A. Related Work

**Bayesian inference for neural networks.** Bayesian deep learning provides a principled framework for uncertainty quantification by maintaining distributions over network weights rather than point estimates (Papamarkou et al., 2024). Several approximate inference families have been developed: The Laplace approximation fits a Gaussian at the MAP estimate (Daxberger et al., 2021); variational inference optimizes a tractable surrogate distribution (Osawa et al., 2019; Shen et al., 2024); and sampling-based methods draw from the posterior via MCMC. Among the latter, stochastic gradient Langevin dynamics (SGLD; Welling & Teh 2011) and stochastic gradient Hamiltonian Monte Carlo (SGHMC; Chen et al. 2014; Springenberg et al. 2016) enable mini-batch posterior sampling, with Ma et al. (2015) providing a unifying framework for the full class of SG-MCMC dynamics. More recently, microcanonical Langevin Monte Carlo (Robnik & Seljak, 2023) and its variants MILE (Sommer et al., 2025) and SMILE (Sommer et al., 2026) have demonstrated state-of-the-art performance for Bayesian neural networks at scale, matching or exceeding other approaches at reduced computational cost. Deep ensembles (Lakshminarayanan et al., 2017) offer a simple but effective alternative: training $K$ independently initialized networks and averaging predictions. Fort et al. (2019) showed that random initializations explore different loss basins, explaining ensemble diversity. The relationship between deep ensembles and Bayesian inference has been further studied by D'Angelo & Fortuin (2021), who showed that adding repulsive terms makes ensemble training equivalent to Bayesian inference via Wasserstein gradient flow.

**PFN foundations and theory.** Müller et al. (2021) introduced PFNs as transformers trained on synthetic data to approximate Bayesian posterior predictive distributions via in-context learning, with Nagler (2023) providing statistical foundations by showing that PFNs admit a frequentist ERM interpretation. This line of work has since catalysed a rapidly growing family of tabular foundation models. On the TabPFN side, TabPFNv2 (Hollmann et al., 2025) and TabPFN-2.5 (Grinsztajn et al., 2025) scale the architecture to larger context windows and richer causal synthetic priors. TabICL (Qu et al., 2025; 2026) provides a fully open-source alternative with strong scaling to large datasets. TabDPT (Ma et al., 2024) proposes the combination of ICL-based retrieval with self-supervised learning on real-world datasets. More recent proposals push the prior-design frontier further: Mitra (Zhang et al., 2025a) combines multiple heterogeneous synthetic priors to improve coverage of real-world distributions, while Limix (Zhang et al., 2025b) extends the in-context learning paradigm to a broader family of structured data beyond canonical tabular format. Taken together, these works establish in-context learning over synthetic priors as the dominant paradigm for tabular foundation models; our analysis of optimization uncertainty and its mitigation via probabilistic training applies to many models in this family.

**Uncertainty quantification in PFNs** Uncertainty in PFN predictions arises at multiple levels, and recent works address different layers of this hierarchy. At the *Context level*, uncertainty stems from the finite downstream dataset $\mathcal{D}_n$: Even a perfect Bayesian predictor is uncertain about the data-generating process $p_0$ when $n$ is small, and this decomposes into *aleatoric* uncertainty (irreducible noise in $y$ given $x$ and $p_0$) and *context-epistemic* uncertainty (reducible uncertainty about $p_0$ given $\mathcal{D}_n$). Two recent works provide principled frameworks for quantifying context-level uncertainty while treating the PFN as a fixed black box. Nagler & Rügamer (2025) construct Bayesian posteriors for PFN estimates via martingale posterior sampling: They treat the PFN's posterior predictive as an informed starting point and iteratively sample synthetic observations, updating the predictive CDF through a Gaussian copula mechanism. Fortini et al. (2026) work within the Bayesian Predictive Inference (BPI) framework: They track how the PFN's one-step-ahead predictive distributions change as context is added, and prove a predictive CLT under quasi-martingale conditions, yielding an explicit aleatoric/epistemic decomposition. Both methods are lightweight (only forward passes through the frozen PFN) and report useful credible intervals with near-nominal coverage.

Against this background, Fortini et al. (2026) argues that weight-space approaches to PFN uncertainty constitute a "category error", as they quantify the uncertainty in the transformers *meta-parameters* and not for a given task at hand. We believe that this characterization is too strong, because it conflates two conceptually distinct and orthogonal levels of uncertainty. *Context uncertainty*, the subject of both frameworks above, arises conditional on a fixed predictor: Given the PFN weights $\boldsymbol{\theta}$ and a finite downstream dataset $\mathcal{D}_n$, how uncertain is the prediction about the data-generating process $p_0$? *Amortization uncertainty*, by contrast, concerns the weights $\boldsymbol{\theta}$ themselves: Since pretraining is stochastic optimization on a non-convex objective, different random seeds and training batches yield functionally distinct predictors, and it is entirely well-posed to ask how much the resulting PFN varies across plausible training outcomes. Bayesian inference over network weights is the principled framework for precisely this second question, placing a prior over $\boldsymbol{\theta}$, forming the posterior $p_\beta(\boldsymbol{\theta} \mid \mathcal{T}_M)$, and marginalizing via Equation (9). Crucially, epistemic uncertainty in the amortization/pretraining process is invisible to any procedure conditioned on a single frozen model: Context-level methods cannot register disagreement between independently

trained PFNs as uncertainty, because that signal lies outside their conditioning set. Our results in Sections 3.1 and E show this is consequential: Cross-warmstart disagreement is measurable and persistent across both scaling axes. A complete uncertainty account for a deployed PFN therefore, requires both levels: Context-level decomposition conditional on a given PFN, and amortization uncertainty over plausible training outcomes, with our contribution addressing the second.

**Amortized/Simulation-based Inference**    PFN pretraining is closely related to amortized simulation-based inference (SBI), where neural density estimators are trained on simulator-drawn samples to bypass intractable likelihoods (Cranmer et al., 2020). Despite differences in scale and downstream task, SBI shares all three sources of uncertainty introduced in Section 2.2. Amortization error has been studied via systematic benchmarking (Lueckmann et al., 2021), demonstrating that it is a persistent property of modern neural posterior and likelihood estimators. The *simulation gap* between simulator and reality, the SBI analogue of our prior error, has received particular attention: Schmitt et al. (2023) proposes a deployment-time misspecification detector, while Kelly et al. (2023) and O'Callaghan et al. (2025) develop estimators that remain robust under simulator misspecification. These ideas are natural candidates for transfer to PFNs, but to our knowledge, no analogue has been developed in the PFN literature.

**Prior misspecification**    Several works in the PFN literature acknowledge that the synthetic prior $\Pi$ is unlikely to fully cover real-world tabular distributions but stop short of measuring the resulting bias for a concrete dataset. Already Hollmann et al. (2025); Qu et al. (2025) report ablations showing strong sensitivity of downstream performance to prior design, which is a direct symptom of prior misspecification, and Müller et al. (2025) explicitly identifies prior design as the central open problem for the PFN paradigm. Subsequent works respond by enriching or adapting the prior: Mitra (Zhang et al., 2025a) mixes multiple synthetic generators to broaden coverage, Drift-Resilient TabPFN (Helli et al., 2024) extends the prior with explicit distribution-shift mechanisms, and Real-TabPFN (Garg et al., 2025) continues pretraining on curated real-world data to close the synthetic-to-real gap. Also, when applied to causal inference, Ma et al. (2025) shows that a misspecified PFN prior can violate identifiability for causal estimands even with infinite context. Despite this widespread awareness, none of these works provide a procedure to quantify, for a given downstream dataset, how much of the predictive error is attributable to prior misspecification.

## B. Limitations and future work

We study models up to $\sim$14M parameters; extending to full-scale TabPFNv2/TabICLv2 is a natural next step, as well as extending it to regression. Also, the conditions under which sampling and possibly other approximate Bayesian inference methods improve upon DEs are under investigation, as they could present a more cost-effective alternative for obtaining diverse samples from the posterior. Furthermore, the extent to which context uncertainty, amortization uncertainty and prior uncertainty are present and relevant for any given prediction task remains to be explored. Likely, amortization uncertainty is the most relevant factor for further improving the overall predictive performance of PFNs, while context uncertainty and prior uncertainty might be what matters the most for informed decision making based on the predictions of a PFN and hence should be studied in more detail. Finally, combining training-time (BDE) and existing inference-time ensembling could address weight-space and dataset-induced variability jointly.

# C. Extended Results

## C.1. Scaling Behaviour results

*Table 2.* Optimization uncertainty on Synthetic datasets. Higher values indicate greater functional disagreement between trained models.

| Model | $OU_{\text{JSD}}$ | $OU_{\text{Dis}}$ | $OU_{\text{Var}}$ | $OU_{\text{NLL}}$ |
|---|---|---|---|---|
| NanoTabICL small | 0.0025 | 0.0362 | 0.0173 | 0.0004 |
| NanoTabICL medium | 0.0032 | 0.0455 | 0.0196 | 0.0024 |
| NanoTabICL large | 0.0018 | 0.0381 | — | — |
| NanoTabPFN small | 0.0041 | 0.0509 | 0.0223 | 0.0056 |
| NanoTabPFN medium | 0.0025 | 0.0440 | 0.0177 | 0.0017 |
| NanoTabPFN large | 0.0016 | 0.0430 | — | — |

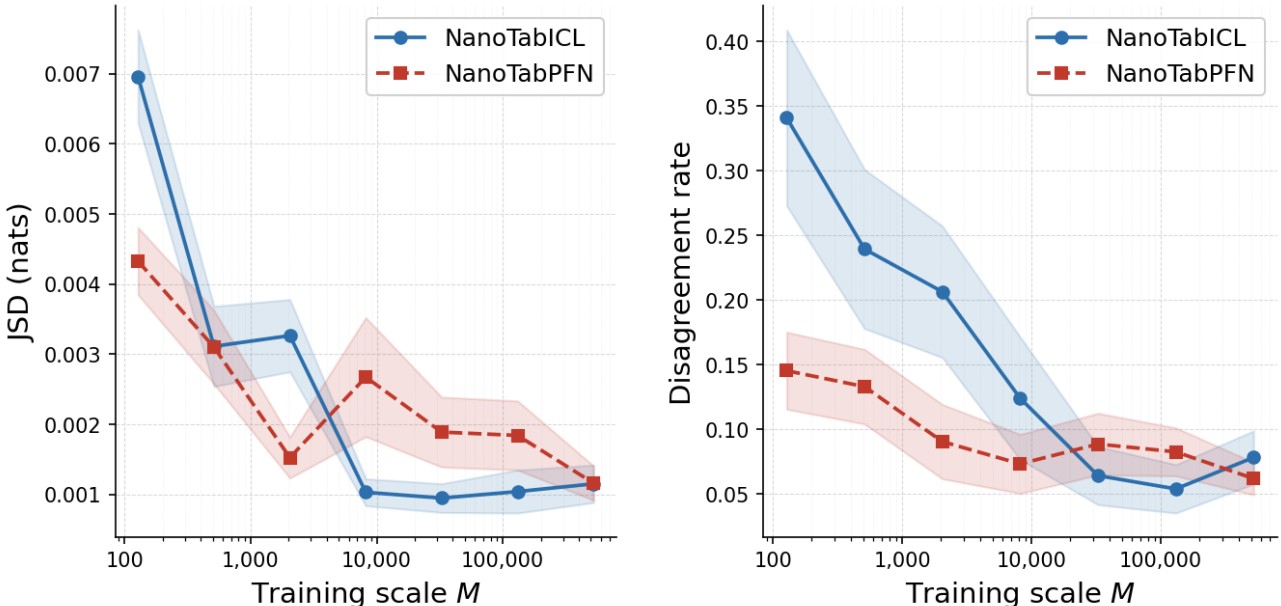

*Figure 2.* Optimization uncertainty as a function of meta-trainingscale M for NanoTabICL and NanoTabPFN (medium) evaluated on TabArena datasets. **Left**: Jensen–Shannon divergence. **Right**: Pairwise disagreement rate. Both metrics decrease at first but plateau at a non-zero level.

## C.2. (Bayesian) Deep Ensembles

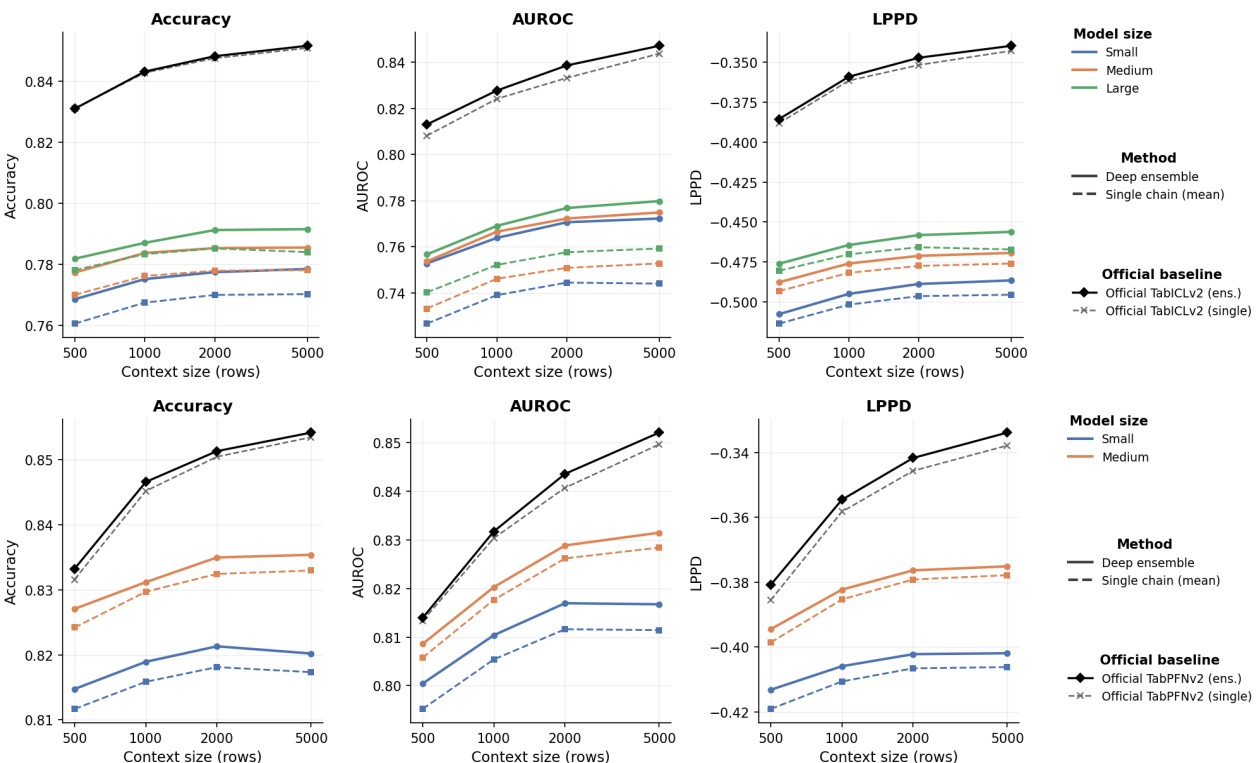

*Figure 3.* Predictive performance of TabICL (**top**) and TabPFN (**bottom**) deep ensembles at context sizes between 500 and 5000 on TabArena datasets. Dashed lines: average single-member performance; solid lines: deep ensemble. Ensemble sizes – TabICL: small=8, medium=8, large=4; TabPFN: small=8, medium=4.

| Small | | Medium | | Large | |
|---|---|---|---|---|---|
| **Model** | Elo | **Model** | Elo | **Model** | Elo |
| TabICLv2 | 1601 | TabICLv2 | 1602 | TabICLv2 | 1599 |
| TabPFNv2.6 | 1591 | TabPFNv2.6 | 1593 | TabPFNv2.6 | 1590 |
| RandomForest | 1000 | **TabPFN–DE-(4)** | **1023** | RandomForest | 1000 |
| **TabPFN–DE-(8)** | **907** | RandomForest | 1000 | Linear | 898 |
| Linear | 898 | **TabPFN–Single** | **972** | **TabICL–DE-(4)** | **760** |
| **TabPFN–Single** | **870** | Linear | 898 | **TabICL–Single** | **743** |
| **TabICL–DE-(8)** | **733** | **TabICL–DE-(8)** | **760** | KNN | 602 |
| **TabICL–Single** | **689** | **TabICL–Single** | **686** | | |
| KNN | 610 | KNN | 600 | | |

*Table 3.* Elo metrics from the official TabArena-light evaluation protocol for the three model sizes together with scores for reference models. Small variations in Elo between model sizes are expected as they are computed by all pairwise comparisons with all *participating* models. The numbers in brackets indicate the number of ensemble members for the deep ensembles.

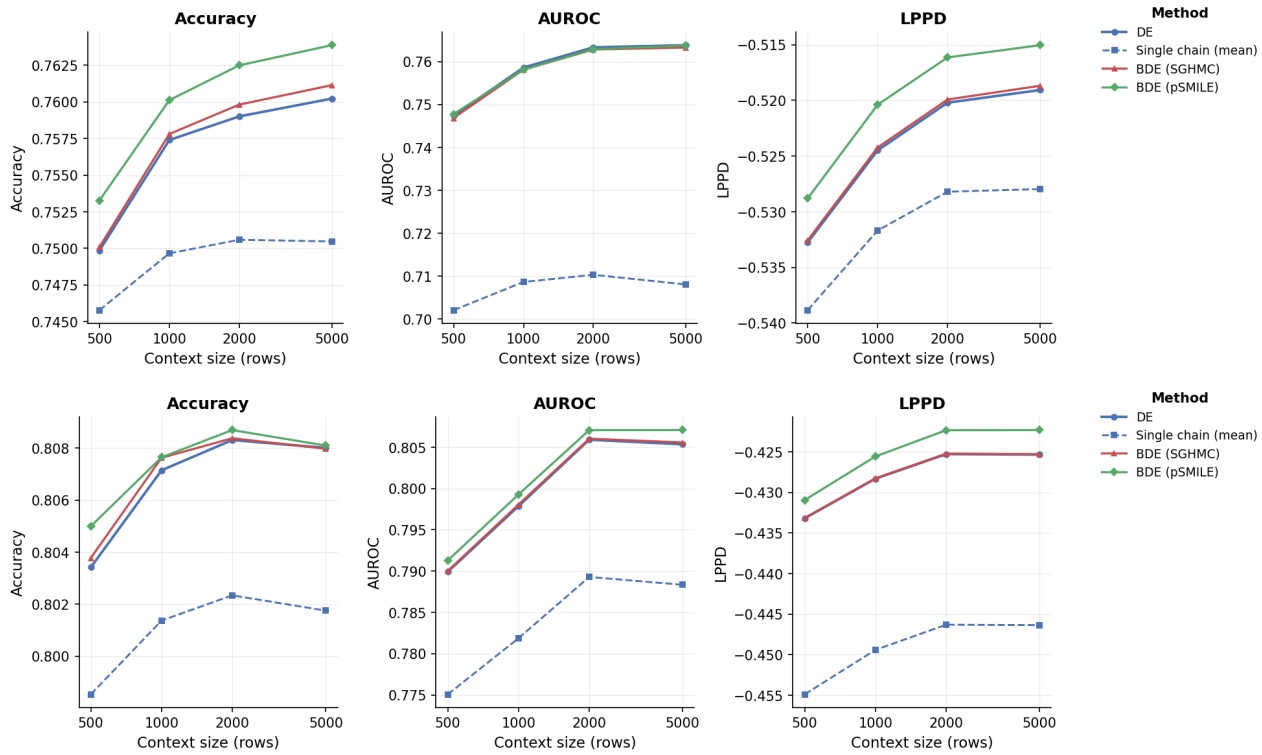

*Figure 4.* Predictive performance of TabICL (**top**) and TabPFN (**bottom**) deep ensembles vs. Bayesian deep ensembles at context sizes between 500 and 5000 on TabArena datasets. Dashed lines: average single-member performance; solid lines: deep ensemble. Ensemble sizes: small=8, medium=4, large=2. The SGHMC-based BDE has 80 members, the pSMILE-based has 160.

*Table 4.* Deep-ensemble results on the TabArena test datasets (up to 5-class classification, 500 context rows). For each model size we report the average single-chain metric (Single avg.) and the deep-ensemble metric (DE).

| Model | TabICL | | | TabPFN | | |
|---|---|---|---|---|---|---|
| | **ACC** | **AUC** | **LPPD** | **ACC** | **AUC** | **LPPD** |
| | *Small (∼2M), DE-8* | | | *Small (∼2M), DE-8* | | |
| Single (avg.) | 0.7606 | 0.7266 | -0.5137 | 0.8117 | 0.7952 | -0.4192 |
| DE | 0.7686 | 0.7528 | -0.5077 | 0.8147 | 0.8004 | -0.4132 |
| | *Medium (∼7M), DE-8* | | | *Medium (∼5M), DE-4* | | |
| Single (avg.) | 0.7701 | 0.7332 | -0.4933 | 0.8242 | 0.8058 | -0.3986 |
| DE | 0.7774 | 0.7536 | -0.4878 | 0.8271 | 0.8086 | -0.3945 |
| | *Large (∼14M), DE-4* | | | | | |
| Single (avg.) | 0.7782 | 0.7402 | -0.4805 | — | — | — |
| DE | 0.7819 | 0.7566 | -0.4761 | — | — | — |

*Table 5.* Results on the synthetic test set (up to 5-class classification, 500 context rows).

| Model | TabICL | | | TabPFN | | |
|---|---|---|---|---|---|---|
| | ACC | AUC | LPPD | ACC | AUC | LPPD |
| | *Small (∼2M), DE-8* | | | *Small (∼2M), DE-8* | | |
| Single (avg.) | 0.7152 | 0.9298 | -0.6293 | 0.7139 | 0.9291 | -0.6359 |
| DE | 0.7185 | 0.9316 | -0.6209 | 0.7174 | 0.9312 | -0.6242 |
| | *Medium (∼7M), DE-8* | | | *Medium (∼5M), DE-4* | | |
| Single (avg.) | 0.7201 | 0.9324 | -0.6171 | 0.7214 | 0.9332 | -0.6141 |
| DE | 0.7213 | 0.9331 | -0.6140 | 0.7224 | 0.9339 | -0.6106 |
| | *Large (∼14M), DE-4* | | | | | |
| Single (avg.) | 0.7216 | 0.9332 | -0.6135 | — | — | — |
| DE | 0.7223 | 0.9337 | -0.6112 | — | — | — |

# D. Optimization Uncertainty

## D.1. Definitions

We define four metrics for quantifying optimization uncertainty across $K$ independently trained warmstarts $\{\boldsymbol{\theta}^{(k)}\}_{k=1}^{K}$, evaluated on $N$ test datasets. All metrics use identical optimizer settings; only random seeds and training data differ.

**Jensen–Shannon Divergence (JSD).**   The generalized JSD (Lin, 1991) decomposes the predictive entropy of the ensemble:

$$H(\bar{q}) = \underbrace{\mathrm{JSD}(q_{\boldsymbol{\theta}^{(1)}}, \ldots, q_{\boldsymbol{\theta}^{(K)}})}_{\text{optimization uncertainty}} + \frac{1}{K} \sum_{k} H(q_{\boldsymbol{\theta}^{(k)}}), \tag{10}$$

where $\bar{q} = \frac{1}{K} \sum_{k} q_{\boldsymbol{\theta}^{(k)}}$. We report the average over test examples:

$$\widehat{OU}_{\mathrm{JSD}} = \frac{1}{N} \sum_{n=1}^{N} \mathrm{JSD}(q_{\boldsymbol{\theta}^{(1)}}(\cdot \mid x_n), \ldots, q_{\boldsymbol{\theta}^{(K)}}(\cdot \mid x_n)). \tag{11}$$

The ratio $\widehat{OU}_{\mathrm{JSD}}/H(\bar{q})$ expresses optimization uncertainty as a fraction of total predictive uncertainty. With equal weights, the JSD equals the mutual information between the prediction and the model index.

**Pairwise disagreement rate.**

$$\widehat{OU}_{\mathrm{dis}} = \frac{1}{\binom{K}{2}} \sum_{i<j} \frac{1}{N} \sum_{n=1}^{N} \mathbf{1}[\hat{y}^{(i)}(x_n) \neq \hat{y}^{(j)}(x_n)]. \tag{12}$$

Compared to the JSD, it only registers disagreement when the argmax flips, and hence underestimates functional disagreement.

**Standard deviation of predicted class probabilities.**

$$\widehat{OU}_{\mathrm{var}} = \sqrt{\frac{1}{N} \sum_{n=1}^{N} \frac{1}{C} \sum_{c=1}^{C} \mathrm{Var}_k[q_{\boldsymbol{\theta}^{(k)}}(y{=}c \mid x_n)]}. \tag{13}$$

**Standard deviation of per-warmstart NLL.**

$$\widehat{OU}_{\mathrm{NLL}} = \sqrt{\mathrm{Var}_k[\mathcal{L}_{\mathrm{test}}(\boldsymbol{\theta}^{(k)})]}, \quad \mathcal{L}_{\mathrm{test}}(\boldsymbol{\theta}^{(k)}) = -\frac{1}{N} \sum_{n=1}^{N} \log q_{\boldsymbol{\theta}^{(k)}}(y_n \mid x_n). \tag{14}$$

## D.2. Optimization uncertainty across model scales

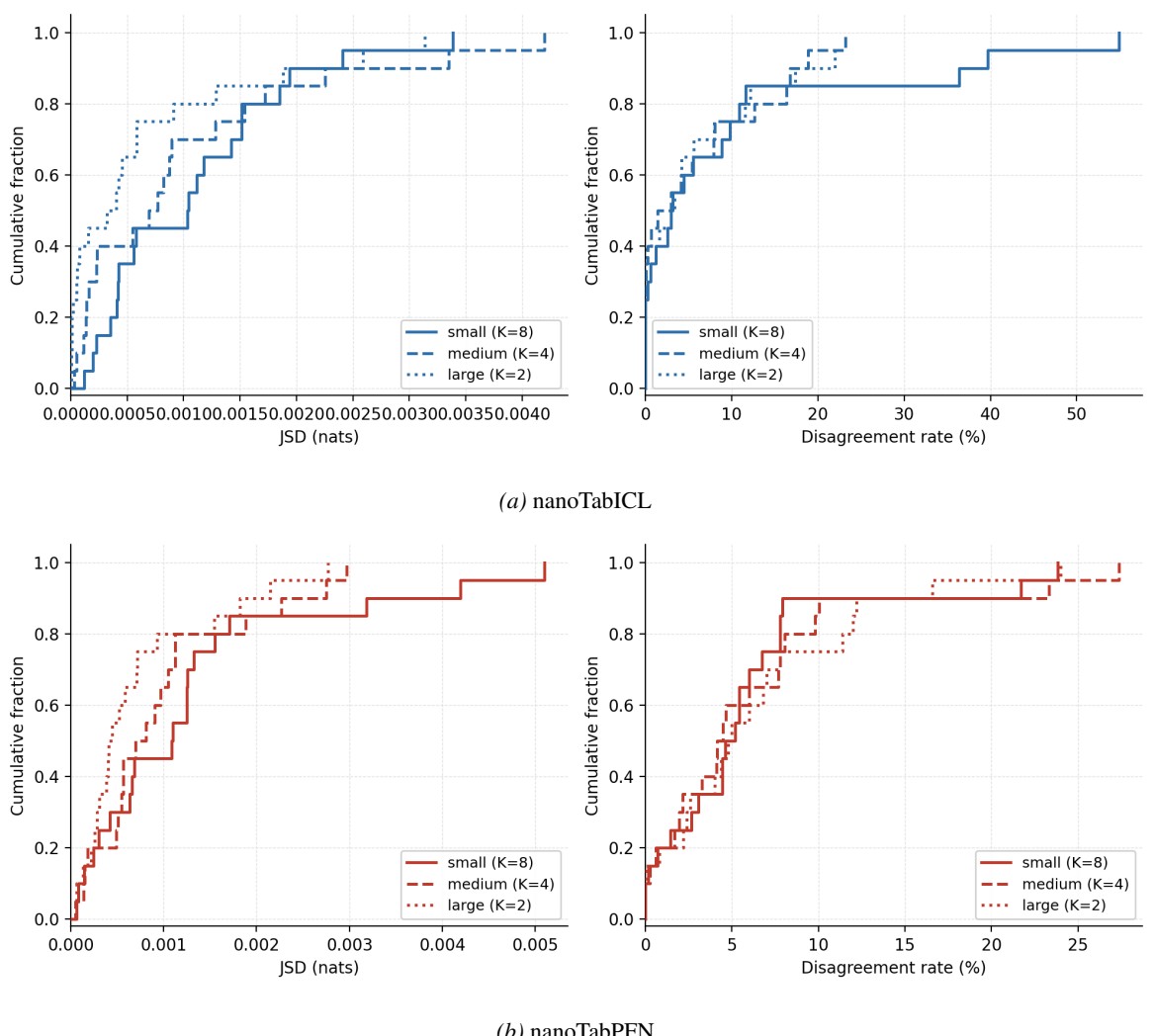

*(a)* nanoTabICL

*(b)* nanoTabPFN

*Figure 5.* Empirical CDFs of per-dataset optimization uncertainty on TabArena. **Left**: JSD (nats), **right**: Pairwise disagreement rate. Computed from K independent warmstarts. Solid lines: small scale (K=8); dashed lines: medium scale (K=4); dotted lines: large scale (K=2). (a) NanoTabICL. (b) NanoTabPFN. The heavy right tail indicates that a non-trivial subset of datasets exhibits substantially higher disagreement than the aggregate mean suggests.

# E. Estimation Uncertainty

The estimator $\widehat{EE}(M)$ introduced in Equation (7) relies on a high-$M$ reference checkpoint $\boldsymbol{\theta}_{\mathrm{ref}}$ and averages over $K$ warmstarts to cancel zero-mean optimization noise. We fit $\log \widehat{EE}(M) = \log C - \alpha \log M$ via linear regression; the fitted lines and data points are reported below.

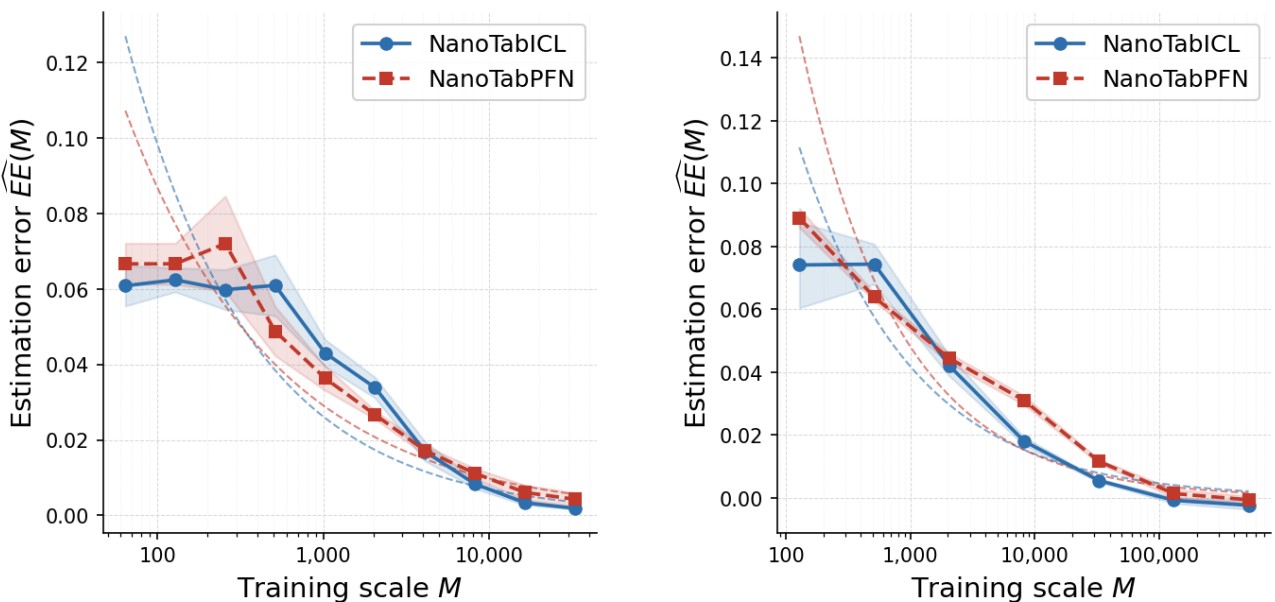

*Figure 6.* Estimation error scaling. **Left**: small architectures, **right**: medium architectures, Estimation error $\widehat{EE}(M)$ vs. meta-training scale $M$, with power-law fit (dashed).

# F. Benchmark datasets and evaluation protocol

**Dataset selection and metric computation.** We evaluate on two real-world tabular benchmarks subject to a shared inclusion filter designed to match the scope of our models. From **TabZilla** (McElfresh et al., 2023) we use the 36-dataset "Hard" suite as a validation dataset during the pretraining. From **TabArena** (Erickson et al., 2025) we use the classification tasks of version v0.1. Both sets are filtered identically: We retain only supervised classification tasks with at most 50 features and 5 classes, no missing values, and a minority-class frequency of at least 2.5%. After this selection process, 15 TabZilla datasets and 20 TabArena datasets remained (see Table 6 below for a list of the dataset names). Datasets exceeding the required maximum number of samples are stratified-subsampled. Performance is measured by accuracy (ACC), macro-averaged one-vs-rest ROC AUC, and log pointwise predictive density. All three metrics are computed on predictions concatenated across one run of stratified 5-fold cross-validation.

| TabArena | Tabzilla |
|---|---|
| Amazon_employee_access | Australian |
| Bank_Customer_Churn | GesturePhaseSegmentationProcessed |
| E-CommereShippingData | ada_agnostic |
| Is-this-a-good-customer | airlines |
| SDSS17 | balance-scale |
| bank-marketing | credit-g |
| blood-transfusion-service-center | electricity |
| churn | elevators |
| credit-g | jungle_chess_2pcs_raw_endgame_complete |
| credit_card_clients_default | kc1 |
| diabetes | monks-problems-2 |
| hazelnut-spread-contaminant-detection | phoneme |
| heloc | qsar-biodeg |
| in_vehicle_coupon_recommendation | socmob |
| maternal_health_risk | vehicle |
| online_shoppers_intention | |
| qsar-biodeg | |
| seismic-bumps | |
| students_dropout_and_academic_success | |
| website_phishing | |

*Table 6.* Datasets used for evaluating real-world performance of PFNs from TabArena and Tabzilla.

# G. Methods

## G.1. PFN architectures

We pretrain three architecture sizes for both nanoTabPFN and nanoTabICL, referred to as *small*, *medium* and *large*. The architectures follow the original designs of Pfefferle et al. (2025) and Qu et al. (2026). The smaller variants reduce the number of layers, embedding dimension, and number of attention heads relative to the published large-scale checkpoints to make repeated pretraining and MCMC sampling feasible within our compute budget. Table 7 and Table 8 report the concrete architecture hyperparameters used in this work.

nanoTabPFN follows a standard transformer encoder applied to a per-cell token representation, so its size is fully described by the embedding dimension, the number of attention heads, the hidden dimension of the MLP block, and the number of stacked layers. nanoTabICL instead uses three stacked attention stages: Column attention, row attention, and the ICL attention block, each with its own number of blocks and number of heads, together with a shared embedding dimension and a small set of structural hyperparameters (feature group size, learnable CLS column tokens, and inducing vectors used in the column attention). The MLP hidden dimension inside each TabICL block is fixed to four times the embedding dimension, following the original TabICL implementation.

*Table 7.* Architecture hyperparameters for the nanoTabPFN models.

| Parameter | TabPFN small | TabPFN medium | TabPFN large |
| --- | --- | --- | --- |
| Number of layers | 4 | 5 | 6 |
| Embedding dimension | 192 | 280 | 384 |
| Number of attention heads | 8 | 8 | 8 |
| MLP hidden dimension | 384 | 560 | 768 |
| Number of output classes | 5 | 5 | 5 |

*Table 8.* Architecture hyperparameters for the nanoTabICL models. Each model contains three stages – column, row, and ICL attention – each with its own depth and number of heads.

| Parameter | TabICL small | TabICL medium | TabICL large |
| --- | --- | --- | --- |
| Embedding dimension | 64 | 128 | 128 |
| Column attention blocks | 3 | 3 | 3 |
| Row attention blocks | 3 | 3 | 3 |
| ICL attention blocks | 5 | 5 | 6 |
| Column attention heads | 4 | 8 | 8 |
| Row attention heads | 4 | 8 | 8 |
| ICL attention heads | 4 | 8 | 8 |
| Feature group size | 3 | 3 | 3 |
| CLS column tokens | 3 | 3 | 4 |
| Column inducing vectors | 64 | 128 | 128 |
| Maximum classes | 5 | 5 | 5 |

## G.2. Optimizer Configuration

The optimizer and its hyperparameters were selected based on a combination of existing literature and hyperparameter tuning. In a first step, a random search with 200 runs on a smaller architecture (separately for nanoTabPFN and nanoTabICL) was used to determine the weight decay and learning rate scheduler. Subsequently, the step size and Scheduler settings were further optimized per model size using a small grid search. Furthermore, for nanoTabPFN for medium and large architectures, a small linear warmup phase was added after initial experiments, which is in line with recommendations from Hollmann et al. (2025). The number of warmstarts per architecture where determined based on the available compute resources.

*Table 9.* Search strategy for hyperparameters

| Parameter | Strategy | Range |
|---|---|---|
| Optimizer | Literature | — |
| $\beta_1$ | Literature | — |
| $\beta_2$ | Literature | — |
| Batch size | Literature | — |
| Weight decay | Random-search | [0.000001, 0.1] |
| Learning rate scheduler | Random-search | [Linear, Cosine] |
| Scheduler end value (fraction) | Grid-Search | [0.01, 0.2] |
| Scheduler decay step (fraction) | Grid-Search | [0.3, 0.5] |
| Initial learning rate | Grid-Search | [0.00005, 0.0005] |

*Table 10.* Optimizer settings for nanoTabICL models

| Parameter | TabICL small | TabICL medium | TabICL large |
|---|---|---|---|
| Optimizer | AdamW | AdamW | AdamW |
| $\beta_1$ | 0.9 | 0.9 | 0.9 |
| $\beta_2$ | 0.99 | 0.99 | 0.99 |
| Weight decay $\beta$ | 0.0001 | 0.0001 | 0.0001 |
| Learning rate scheduler | Cosine | Cosine | Cosine |
| Initial learning rate | 0.0004 | 0.0004 | 0.0004 |
| Batch size | 64 | 64 | 64 |
| Number of datasets | 0.96M | 2.496M | 4.096M |
| Number of warmstarts | 8 | 4 | 2 |

*Table 11.* Optimizer settings for nanoTabPFN models

| Parameter | TabPFN small | TabPFN medium | TabPFN large |
|---|---|---|---|
| Optimizer | AdamW | AdamW | AdamW |
| $\beta_1$ | 0.9 | 0.9 | 0.9 |
| $\beta_2$ | 0.99 | 0.99 | 0.99 |
| Weight decay $\beta$ | 0.0001 | 0.0001 | 0.0001 |
| Learning rate scheduler | Cosine | Cosine + Linear Warmup | Cosine + Linear Warmup |
| Initial learning rate | 0.0004 | 0.0001 | 0.0001 |
| Batch size | 64 | 64 | 64 |
| Number of datasets | 0.96M | 2.496M | 4.096M |
| Number of warmstarts | 8 | 4 | 2 |

## G.3. Sampler Configuration

MCMC sampling is initialized from independently warm-started checkpoints as proposed in Sommer et al. (2024). We run 8 parallel chains for the small architecture. Throughout all experiments, a batch size of 64 datasets is used with potential micro-batching to address GPU memory constraints. Per sampler, a grid search was used to find the best performing hyperparameters, with special focus on the interplay of step size and the temperature $\beta$ as defined in Equation (8). Table 12 reports the best performing sampler configurations for the nanoTabICL experiments and Table 13 reports the best found configurations for nanoTabPFN. For cSGLD, even though an extensive grid search over all relevant hyperparameters was conducted, no settings could be found that performed on par with with the deep ensemble. For this reason, it is exluded from the tables below.

*Table 12.* MCMC sampler hyperparameters for the TabICL small experiments.

| Parameter | adaSGHMC | pSMILE |
|---|---|---|
| Step size | 0.005 | 0.005 |
| Warmup steps | 500 | 500 |
| Sampling steps | 2500 | 2500 |
| Thinning interval | 250 | 125 |
| Temperature $\beta$ | 1.0 | 1.0 |
| Batch size (sampling) | 64 | 64 |
| Chains | 8 | 8 |
| Prior $\sigma_{\boldsymbol{\theta}}$ | 1.0 | 1.0 |

*Table 13.* MCMC sampler hyperparameters for the TabPFN small experiments.

| Parameter | adaSGHMC | pSMILE |
|---|---|---|
| Step size | 0.005 | 0.0005 |
| Warmup steps | 500 | 2000 |
| Sampling steps | 1000 | 8000 |
| Thinning interval | 100 | 400 |
| Temperature $\beta$ | 1.0 | 1.0 |
| Batch size (sampling) | 64 | 64 |
| Chains | 8 | 8 |
| Prior $\sigma_{\boldsymbol{\theta}}$ | 1.0 | 1.0 |

### G.4. Software and computing environment

Experiments were implemented in Python using `jax` (Bradbury et al., 2018), `BlackJAX` (Cabezas et al., 2024) and extensions of the codebase of (Sommer et al., 2025; 2026). The `pytorch` implementations of nanoTabPFN (Pfefferle et al., 2025) and TabICL (Qu et al., 2026) were used as a starting point and subsequently ported to `jax`. The experiments were conducted on two NVIDIA RTX A6000 GPUs (with 48 GB VRAM each) and AMD Ryzen™ Threadripper™ PRO 5000WX/3000WX CPU with 64 cores. Additionally, larger trainings were performed with two additional Nvidia H100 GPUs (with 94 GB VRAM each).

# H. Ablation Studies

## H.1. Ensemble Size

We evaluate how the deep ensemble's predictive quality scales with the number of warmstarts $K$ on the synthetic test set. For this, LPPD is used as it is a proper scoring rule that simultaneously rewards calibration and sharpness. Figure 7 shows the sequential-chain LPPD curve. Starting from $K=1$, each point is the LPPD obtained by averaging the predictive distributions of the first $K$ warmstarts. To remove any bias from the arbitrary order in which the ensemble members are added, we repeat this procedure over ten random orderings of the full chain pool and report the mean together with a $\pm 1\,\sigma$ band across orderings.

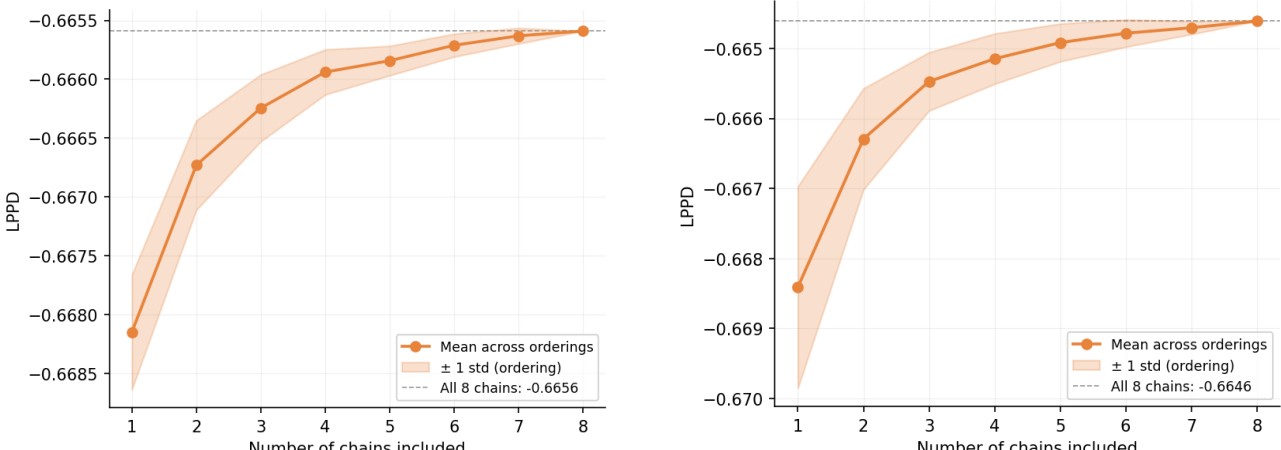

*Figure 7.* **Left:** nanoTabICL (2M parameters), **right:** nanoTabPFN (2M parameters). Sequential-chain LPPD on the synthetic test set as a function of ensemble size $K$. The shaded band shows $\pm 1\,\sigma$ over ten random chain orderings; it narrows as $K$ grows and collapses to zero when all chains are included.

Both models show the same qualitative pattern: LPPD improves sharply from $K=1$ to $K=4$ and then saturates, with the ordering uncertainty also decreasing substantially. The large jump at $K=2$ confirms that even two diverse warmstarts capture a significant portion of the functional diversity available in the ensemble. In practice, more warmstarts are always preferable as they monotonically increase LPPD but since each warmstart requires a full meta-training run, budget constraints often limit $K$. The results suggest that *any* ensemble of $K \geq 2$ warmstarts is clearly preferable to a single model, with further gains for more warmstarts, if compute permits. Similar patterns with a relatively modest number of warm-started chains saturating predictive performance have also been found in existing literature (Kobialka et al., 2026; Sommer et al., 2025).

