# OpenReview forum: "On the Uncertainty in Prior-Data Fitted Network Pretraining"
_ICML.cc/2026/Workshop/FMSD — FMSD @ ICML 2026 Poster_

### Official Review · Reviewer_MNzC · 2026-05-21
**Insightful error decomposition and scaling analysis**

**Rating:** 7
**Confidence:** 4

**Review:**

The paper investigates the pretraining and amortization uncertainty of priod-data fitted networks.
The authors first formalize a framework that decomposes PFN error into: prior error (mismatch between test task and training prior), context error (information gap withing the current task's context) and amortization error (mismatch between the training objective and the actual trained model). They then focus on the amortization error and provide the classic ERM error decomposition into: approximation, estimation and optimization. The show that scaling the training data or model size does not consistently improve the optimization error. Finally, they propose to train the model using a Bayesian ensemble method to mitigate the optimization error.

Strengths:

1. The question they study regarding untangling the uncertainty factors in PFN provides an interesting framework for the community which to followed up on and expand.
2. The results regarding scaling the training data and model size and its effect on the optimization error is insightful and interesting to the community.
3. The paper is a good fit for the workshop.

Areas for improvement:

I find the proposed method to mitigate the optimization error not convincing enough. I don't think this is required for the workshop but for future versions the empirical justification for this should be extended and show more significant results.


Justification of Score:

This paper is well-suited for the workshop. While the practical ensembling solutions are not convincingly significant, the theoretical error decomposition and the empirical scaling experiments provide interesting insights that are valuable to the community of PFN research.

---

### Official Review · Reviewer_7XZP · 2026-05-21
**The paper provides a formal investigation into the pretraining-time uncertainties of Prior-Data Fitted Networks (PFNs)**

**Rating:** 6
**Confidence:** 4

**Review:**

**Strengths**
- tackles uncertainty decomposition from a training-time perspective
- paper is understood well with nice definitions
- nice empirical findings on optimization uncertainty against scaling

**Areas for Improvement**
- How is the "MCMC-Sampling" conducted specifically? Is it applied via LoRA?
- “deep ensembles followed by MCMC-based posterior sampling” -- how much additional compute is needed?

---

### Official Review · Reviewer_5sQE · 2026-05-22
**Promising paper; further experiments required**

**Rating:** 6
**Confidence:** 4

**Review:**

This paper investigates the source of uncertainties in PFNs (Prior-Fitted Networks) during pretraining. First, it discusses that the gap between true and learned PPDs can be attributed to amortization, context, and prior gap. Then, using an empirical risk minimization framework, the paper defines the amortization error as the sum of the optimization, estimation, and approximation errors. It focuses particularly on optimization uncertainty (OU), exploring techniques to empirically approximate this quantity. Two considered approaches are: deep ensemble (independent PFNs training) and stochastic gradient MCMC methods for training PFNs. In summary, the paper concludes that "neither scaling training data nor scaling model capacity drives optimization uncertainty to zero".

The main strength of the paper is that it addresses an important aspect of PFNs that remains underexplored. The paper is well-written and clear; it was a nice read for me. While the paper leverages existing techniques (deep ensemble and MCMC) for assessing uncertainties, the application of these techniques remains novel, with prior work studying uncertainties only at inference time.

Despite the strengths, I have two main comments.

- It is unclear how we can generalize these findings to different optimizer setups. In particular, would it not make sense to first conduct an intensive hyperparameter optimization (at least on learning rate and batch size), then assess if the same conclusion holds? I think this aspect should be investigated further to make the claim even stronger.
- The claim of the paper seems very general to me, as it sounds like it targets the general PFNs framework, while the experiments only show evidence on two architectures. It would be great to include a discussion on how model architecture affects the results. For example, Figure 1 shows that two architectures yield different trends. Even beyond that, would we expect a non-transformer-based model but fitted on synthetic data (same as defined in PFNs) to arrive at the same conclusion?

Overall, the paper addresses an intriguing question and explores a highly promising direction. However, I believe further experiments or discussion are necessary to fully consolidate the findings.